# An online passive-aggressive algorithm for difference-of-squares classification

**Lawrence K. Saul**
Department of Computer Science and Engineering
University of California, San Diego
9500 Gilman Drive, Mail Code 0404
La Jolla, CA 92093-0404
`saul@cs.ucsd.edu`

## Abstract

We investigate a low-rank model of quadratic classification inspired by previous work on factorization machines, polynomial networks, and capsule-based architectures for visual object recognition. The model is parameterized by a pair of affine transformations, and it classifies examples by comparing the magnitudes of vectors that these transformations produce. The model is also *over-parameterized* in the sense that different pairs of affine transformations can describe classifiers with the same decision boundary and confidence scores. We show that such pairs arise from discrete and continuous symmetries of the model's parameter space: in particular, the latter define symmetry groups of rotations and Lorentz transformations, and we use these group structures to devise appropriately invariant procedures for model alignment and averaging. We also leverage the form of the model's decision boundary to derive simple margin-based updates for online learning. Here we explore a strategy of *passive-aggressive learning*: for each example, we compute the minimum change in parameters that is required to predict its correct label with high confidence. We derive these updates by solving a quadratically constrained quadratic program (QCQP); interestingly, this QCQP is *nonconvex but tractable*, and it can be solved efficiently by elementary methods. We highlight the conceptual and practical contributions of this approach. Conceptually, we show that it extends the paradigm of passive-aggressive learning to a larger family of nonlinear models for classification. Practically, we show that these models perform well on large-scale problems in online learning.

## 1 Introduction

As data sets grow in size and complexity, they create new opportunities—and challenges—for large-scale applications of online learning [1]. These challenges have been extensively studied and, for the most part, elegantly resolved for the simplest linear models of classification [2]. For such models, one particularly elegant approach is that of *passive-aggressive learning* [3]. In this framework, a model is only updated when it fails to classify an example correctly with high confidence. When an update is triggered, however, the model is changed by whatever minimum amount is required to achieve this goal. This approach neatly dispenses with the need to choose or adapt learning rates.

Given the appeal of such updates, it is of natural interest to extend this approach to a larger family of nonlinear models. Proceeding from the linear model, this can be done most straightforwardly by the use of kernel methods [4–6]. But kernel methods do not scale effortlessly to the regime of online learning that we envision in this paper—where the examples are arriving in a streaming fashion from an essentially unlimited source [7, 8]. It is possible to adopt a budgeted approach [9–14]

35th Conference on Neural Information Processing Systems (NeurIPS 2021).

for kernel-based passive-aggressive updates [15–17], conserving memory at the expense of model capacity, but this approach necessarily entails further complexities.

Without the kernel trick, we must face the crux of the problem. Passive-aggressive updates hinge on the ability to perform a fundamental calculation: when an update is triggered, we must compute the minimum change in model parameters that is required to fix the classifier's decision boundary. This is a relatively simple calculation for linear models (with hyperplane decision boundaries), but an enormously complex one for (say) decision trees, ensemble methods, and neural nets with threshold or ReLU activation functions. The question is whether this is true for all nonlinear models.

This paper investigates a family of low-rank models for quadratic classification inspired by previous work on factorization machines [18, 19], polynomial networks [20, 21], and capsule-based architectures for visual object recognition [22, 23]. For models in this family, we show how to derive passive-aggressive updates by solving a quadratically constrained quadratic program (QCQP). The QCQP is nonconvex but tractable: it reduces to an elementary minimization over a bounded interval. In this way, we are able to extend the framework of passive-aggressive learning to a larger family of nonlinear models. This is the paper's main contribution.

The organization of this paper is as follows. In section 2, we review the main lines of work that motivated this study. In section 3, we formulate our model, elucidate its symmetries, and derive the updates for passive-aggressive learning. In section 4, we provide experimental results on a data set with 100M training examples. Finally, in section 5, we conclude and suggest some directions for future work.

## 2    Background and related work

Passive-aggressive updates [3] emerged from a long line of work on incremental learning of linear classifiers [24–33]. These updates are based on a simple intuition. Suppose that an algorithm has access to a stream of labeled examples $\{(\mathbf{x}_t, y_t)\}_{t \geq 1}$, where $\mathbf{x}_t \in \mathbb{R}^n$ and $y_t \in \{-1, +1\}$. The goal of the algorithm is to learn a linear classifier parameterized by a weight vector $\mathbf{w} \in \mathbb{R}^n$. At the outset, the weight vector is initialized to the origin. Then, after the algorithm sees each example, the weight vector is updated by solving the following constrained optimization:

$$\mathbf{w}_{t+1} \;=\; \operatorname*{argmin}_{\mathbf{w} \in \mathbb{R}^n} \|\mathbf{w} - \mathbf{w}_t\|^2 \quad \text{such that} \quad y_t \mathbf{w}^\top \mathbf{x}_t \;\geq\; 1. \tag{1}$$

The optimization is trivial if the algorithm already classifies the input $\mathbf{x}_t$ correctly with at least unit margin: in this case, the algorithm responds *passively*, making no change at all to the weight vector. When this is not the case, however, the algorithm responds *aggressively*, changing $\mathbf{w}$ by the minimum amount required to achieve this goal. The optimization in eq. (1) has the closed-form solution:

$$\mathbf{w}_{t+1} \;=\; \mathbf{w}_t + \alpha_t y_t \mathbf{x}_t \quad \text{where} \quad \alpha_t \;=\; \frac{\max(0, 1 - y_t \mathbf{w}^\top \mathbf{x}_t)}{\|\mathbf{x}_t\|^2}. \tag{2}$$

This update is appealing in its simplicity and easily adapted to handle noisy labels or to learn a separating hyperplane that does not pass through the origin [3]. Similar updates have been studied for problems in online regression [3], portfolio selection [34], nonnegative matrix factorization [35], and active learning [36].

The above framework can also be generalized via the "kernel trick" [4–6] to learn nonlinear classifiers. This is done by substituting kernel function evaluations for dot products between training inputs. In this case, however, the kernelized algorithm will typically limit the number of examples that are used to construct the model's decision boundary [9–17].

In this paper we also develop passive-aggressive updates for a nonlinear model of classification. Our approach, however, is not rooted in the use of kernel methods, but in low-rank quadratic models of classification. For practitioners, these models can provide a natural bridge between purely linear and fully quadratic models of classification. Previous examples of such models include so-called factorization machines [18, 19] and polynomial networks (of degree two) [20, 21]. Though not the focus of this paper, it should be noted that factorization machines (and their variants) have been widely applied to problems in collaborative filtering and click-through rate prediction [37–39].

The specific form of our model is also motivated by so-called capsule architectures for visual object recognition [22, 23]. Like traditional neural nets, capsule-based architectures compute vectors of

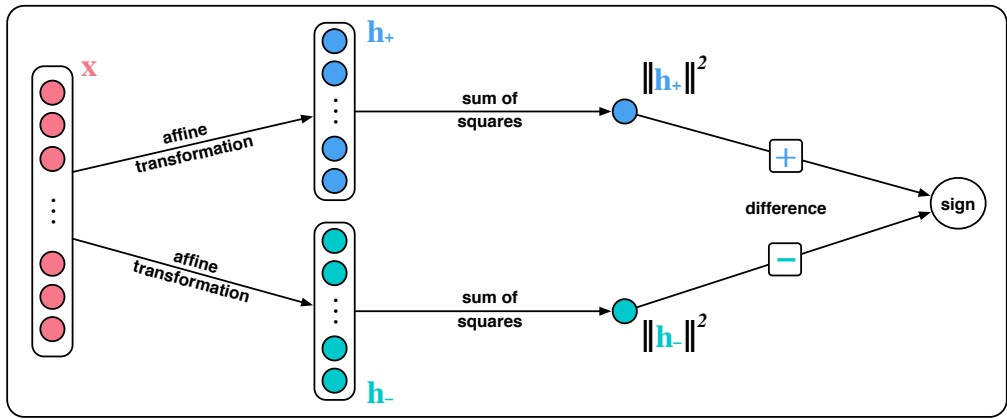

Figure 1: Difference-of-squares (DoS) classifier visualized as a neural network with one layer of hidden units. The network uses a pair of affine transformations to map each input $\mathbf{x}$ into vectors $\mathbf{h}_+$ and $\mathbf{h}_-$ of hidden unit activities. Then it labels the input by comparing the squared magnitudes of these vectors.

hidden unit activities; unlike traditional neural nets, they use the *magnitudes* of these vectors to encode the probability that an entity is present in an image. This idea has mainly been developed for deep architectures with multiple layers of interlinked capsules [23, 40–52], although recently it was explored for a simpler latent variable model of multiway classification [53]. Here we pursue this simplification even further for problems of binary classification. Whereas the latent variable model in [53] is well suited for maximum likelihood estimation in a batch setting, the model in this paper is much better suited to passive-aggressive learning in an online setting. We develop these ideas further in the next section.

## 3 Model

In this section we introduce the model at the heart of this paper. We begin by formulating the model as a neural network (3.1) and elucidating the symmetries of its parameter space (3.2). Next we derive the passive-aggressive updates for online learning (3.3) and show how to average different models across time (3.4).

### 3.1 Difference-of-squares classification

The model we study is most easily visualized as the neural network in Fig. 1. The network maps the input into a hidden layer by a pair of affine transformations: $\mathbf{x} \mapsto \mathbf{A}_\pm \mathbf{x} + \mathbf{b}_\pm$. Then it predicts a binary label $y \in \{-1, +1\}$ by comparing the magnitudes of the vectors produced in this way. More concretely, let $\mathbf{h}_\pm \in \mathbb{R}^d$ denote the vectors of hidden unit activities. The label is predicted as follows:

$$\mathbf{h}_+ = \mathbf{A}_+ \mathbf{x} + \mathbf{b}_+, \tag{3}$$
$$\mathbf{h}_- = \mathbf{A}_- \mathbf{x} + \mathbf{b}_-, \tag{4}$$
$$y = \text{sign}\left(\|\mathbf{h}_+\|^2 - \|\mathbf{h}_-\|^2\right). \tag{5}$$

Based on the form of eq. (5), we refer to this model as a difference-of-squares (DoS) classifier. The dimensionalities of $\mathbf{h}_+$ and $\mathbf{h}_-$ are hyperparameters of the model, which for convenience we set to be equal. Eq. (5) also makes plain the connection to capsule-based architectures [22, 23, 53], with $\mathbf{h}_+$ and $\mathbf{h}_-$ playing the role of hidden "pose" vectors whose squared magnitudes determine how inputs are classified. It should be noted that networks of this form are not universal function approximators because their decision boundary is necessarily quadratic. This is in contrast to neural networks with non-polynomial activation functions, which can approximate any decision boundary in the limit of infinitely many hidden units [54–57].

The DoS model may not be a universal approximator, but with sufficiently many hidden units, it can mimic any fully quadratic classifier. For example, consider a classifier of the form

$$y = \text{sign}\left(\mathbf{x}^\top \mathbf{P} \mathbf{x} + \mathbf{q}^\top \mathbf{x} + r\right), \tag{6}$$

where the decision boundary is determined by the parameters $\mathbf{P} \in \mathbb{R}^{n \times n}$, $\mathbf{q} \in \mathbb{R}^n$ and $r \in \mathbb{R}$. To prove the claim, we simply express $\mathbf{P}$ as the difference of positive semidefinite matrices $\mathbf{P}_+$ and $\mathbf{P}_-$. In particular, if $\mathbf{P} = \mathbf{P}_+ - \mathbf{P}_-$, then eq. (6) can be recast in the form of eqs. (3–5) by setting

$$\mathbf{A}_{\pm} = \left[ \begin{array}{c} \mathbf{q}^{\top} \\ \mathbf{S}_{\pm} \end{array} \right], \quad \mathbf{b}_{\pm} = \left[ \begin{array}{c} r \pm \frac{1}{4} \\ \mathbf{0} \end{array} \right], \tag{7}$$

where $\mathbf{P}_{\pm} = \mathbf{S}_{\pm}^{\top} \mathbf{S}_{\pm}$. From the mapping in eq. (7), we also see the potential of this parameterization. Suppose that the input $\mathbf{x}$ is high dimensional but that the matrix $\mathbf{P}$ is of low rank. Then the matrices $\mathbf{S}_{\pm}$ satisfying $\mathbf{P}_{\pm} = \mathbf{S}_{\pm}^{\top} \mathbf{S}_{\pm}$ can be short and wide as opposed to square, and the matrices $\mathbf{A}_{\pm}$ in eq. (7) will have many fewer rows than columns. Such a DoS model, with so many fewer parameters, may be easier to estimate (robustly) than a fully quadratic classifier.

DoS models differ slightly in form and motivation from the low-rank models of quadratic classification provided by factorization machines [18]. In factorization machines, the decision boundary is modeled by a quadratic surface

$$0 = \sum_{j > i} (P_+)_{ij} x_i x_j + \sum_i q_i x_i + r, \tag{8}$$

where the so-called interaction matrix $\mathbf{P}_+$ is a low-rank positive semidefinite (PSD) matrix that can be expressed as the product of smaller factors. There are two differences between eq. (6) for DoS models and eq. (8) for factorization machines: first, the interaction matrix in factorization machines is restricted to be PSD, and second, the decision boundary in factorization machines is only computed from the off-diagonal terms of this interaction matrix. The first of these differences can be viewed as a purposeful form of regularization for the sorts of extremely sparse prediction problems, such as collaborative filtering and click-through rate prediction, to which factorization machines have been widely applied [19, 21, 37–39]; the second does not have any effect if the input $\mathbf{x}$ is a vector of binary elements (because $x_i^2 = x_i$ if $x_i \in \{0, 1\}$). As shown by eqs. (6–7), however, these restrictions are not shared by DoS classifiers. In the latter, they are overcome by associating separate low-rank matrices (namely, $\mathbf{A}_{\pm}$) to the classes of positive and negative examples.

Before proceeding, we adopt a more unified notation in DoS models for the parameters $\mathbf{A}_{\pm}$ and $\mathbf{b}_{\pm}$. We do so by working in an augmented input space; specifically we define

$$\mathbf{z} = \left[ \begin{array}{c} \mathbf{x} \\ 1 \end{array} \right], \quad \mathbf{U} = [\mathbf{A}_+ \ \mathbf{b}_+], \quad \mathbf{V} = [\mathbf{A}_- \ \mathbf{b}_-], \tag{9}$$

so that the vector $\mathbf{z}$ has exactly one more row than the input $\mathbf{x}$, and the matrices $\mathbf{U}$ and $\mathbf{V}$ have exactly one more column than the matrices $\mathbf{A}_+$ and $\mathbf{A}_-$. In this notation, the model's decision boundary simplifies to

$$y = \text{sign} \left( \|\mathbf{U}\mathbf{z}\|^2 - \|\mathbf{V}\mathbf{z}\|^2 \right) = \text{sign} \left( \mathbf{z}^{\top} \left[ \mathbf{U}^{\top} \mathbf{U} - \mathbf{V}^{\top} \mathbf{V} \right] \mathbf{z} \right). \tag{10}$$

The first expression in eq. (10) provides the most efficient way to compute the label $y \in \{-1, +1\}$, but the second makes plain the model's symmetries. We explore these symmetries in the next section.

## 3.2 Symmetries of the model

Eq. (10) shows that the model's decision boundary and confidence scores only depend on its parameters through the difference $\mathbf{U}^{\top} \mathbf{U} - \mathbf{V}^{\top} \mathbf{V}$. From this observation we deduce two important symmetries. First, the model's predictions are invariant to *orthogonal* transformations of the form

$$\mathbf{U} \mapsto \mathbf{\Omega} \mathbf{U}, \tag{11}$$
$$\mathbf{V} \mapsto \mathbf{\Lambda} \mathbf{V}, \tag{12}$$

where $\mathbf{\Omega}$ and $\mathbf{\Lambda}$ are (independently chosen) $d \times d$ orthogonal matrices, satisfying $\mathbf{\Omega}^{\top} \mathbf{\Omega} = \mathbf{\Lambda}^{\top} \mathbf{\Lambda} = \mathbf{I}$. Second, the model's predictions are invariant to *Lorentz* transformations (or " boosts") of the form

$$\mathbf{U} \mapsto \mathbf{U} \cosh \varphi - \mathbf{V} \sinh \varphi, \tag{13}$$
$$\mathbf{V} \mapsto \mathbf{V} \cosh \varphi - \mathbf{U} \sinh \varphi, \tag{14}$$

where $\varphi \in \mathbb{R}$. (Lorentz transformations are a symmetry of spacetime in the theory of special relativity [58].) Both symmetries reveal that the model is *over-parameterized*; by this we mean

that models with different parameters may describe classifiers with the same decision boundary and confidence scores. This property is commonly observed in deep neural networks with ReLU hidden units [59], and rotational symmetries in particular are characteristic of many models of matrix factorization and low-dimensional embedding [60].

The orthogonal and Lorentz transformations in eqs. (11–14) reflect fundamentally different symmetries of the DoS parameter space. Note in particular that the former do not change the Frobenius norms of the matrices $\mathbf{U}$ and $\mathbf{V}$, while the latter do. From eqs. (13–14) it is possible, in fact, to compute the Lorentz transformation that yields the model parameters of minimum norm:

$$\varphi^* \;=\; \underset{\varphi}{\operatorname{argmin}} \left\{ \left\| \mathbf{U}\cosh\varphi - \mathbf{V}\sinh\varphi \right\|_F^2 + \left\| \mathbf{V}\cosh\varphi - \mathbf{U}\sinh\varphi \right\|_F^2 \right\} \tag{15}$$

As shorthand notation, we define the matrix inner product $\langle \mathbf{U}, \mathbf{V} \rangle = \sum_{ij} U_{ij} V_{ij}$. Then the solution to eq. (15) takes the simple form:

$$\varphi^* \;=\; \frac{1}{2}\tanh^{-1}\left( \frac{2\langle \mathbf{U}, \mathbf{V} \rangle}{\|\mathbf{U}\|_F^2 + \|\mathbf{V}\|_F^2} \right). \tag{16}$$

We omit the derivation of this result, which is obtained by zeroing the derivative of the bracketed expression in eq. (15). The result will be needed in section 3.4, when we discuss how to average different models across time.

### 3.3 Passive-aggressive updates for online learning

We consider learning in the online setting where labeled examples arrive one at a time and are subsequently discarded. Let $(\mathbf{U}_t, \mathbf{V}_t)$ denote the model parameters before the example $(\mathbf{x}_t, y_t)$ arrives at time $t$. We explore a passive-aggressive approach, in which at each time step, the algorithm updates its model by solving the following optimization:

$$\min_{\mathbf{U}, \mathbf{V}} \left\{ \|\mathbf{U} - \mathbf{U}_t\|_F^2 + \|\mathbf{V} - \mathbf{V}_t\|_F^2 \right\} \quad \text{such that} \quad y_t\left( \|\mathbf{U}\mathbf{z}_t\|^2 - \|\mathbf{V}\mathbf{z}_t\|^2 \right) \geq 1. \tag{17}$$

Suppose that the example $(\mathbf{x}_t, y_t)$ is already classified correctly and with high confidence. Then the algorithm does not change the model: i.e., $\mathbf{U}_{t+1} = \mathbf{U}_t$ and $\mathbf{V}_{t+1} = \mathbf{V}_t$. On the other hand, when this is not the case, the algorithm performs the minimal update to correct this failure.

The optimization in eq. (17) is a quadratically constrained quadratic program (QCQP). This QCQP is not convex due to the difference of squared terms that appears in its constraint. It is, however, tractable [61–63]: nonconvex QCQPs with a single constraint can be efficiently solved by a so-called *S-procedure* from nonlinear control theory [64, 65]. This general procedure is beyond the scope of this study, but for the problem of interest in eq. (17), the solution can be obtained and justified by more elementary methods.

We start by presenting the solution as a *fait accompli*. The optimization over matrices $(\mathbf{U}, \mathbf{V})$ in eq. (17) reduces, in the end, to determining a step size $\alpha_t$ analogous to the one that appears in eq. (2). For DoS models, the step size is found by solving the following one-dimensional optimization:

$$\alpha_t \;=\; \underset{\nu \in (0,1)}{\operatorname{argmin}} \left[ \left( \frac{1}{1 - y_t\nu} \right) \|\mathbf{U}_t\mathbf{z}_t\|^2 + \left( \frac{1}{1 + y_t\nu} \right) \|\mathbf{V}_t\mathbf{z}_t\|^2 - \nu \right]. \tag{18}$$

This minimization is convex, and in practice it is easily solved by standard methods (e.g., golden section search). Finally, given the step size $\alpha_t$, the updates[1] for $\mathbf{U}$ and $\mathbf{V}$ take the simple form:

$$\mathbf{U}_{t+1} \;=\; \mathbf{U}_t + \left( \frac{\alpha_t}{y_t - \alpha_t} \right) \cdot \frac{(\mathbf{U}_t\mathbf{z}_t)\mathbf{z}_t^\top}{\|\mathbf{z}_t\|^2}, \tag{19}$$

$$\mathbf{V}_{t+1} \;=\; \mathbf{V}_t - \left( \frac{\alpha_t}{y_t + \alpha_t} \right) \cdot \frac{(\mathbf{V}_t\mathbf{z}_t)\mathbf{z}_t^\top}{\|\mathbf{z}_t\|^2}. \tag{20}$$

We emphasize that these updates are very nearly as simple to implement as the one for linear classifiers in eq. (2). At the same time, however, they provide access to a much richer family of models.

---

[1]These updates assume that neither $\|\mathbf{U}_t\mathbf{z}\|$ nor $\|\mathbf{V}_t\mathbf{z}\|$ is equal to zero. The assumption holds in practice, as such precise cancellations do not occur when the model parameters are initialized with random values.

Next we briefly justify the form of these updates. The calculation that follows is a special case of a more general treatment [66] for nonconvex but tractable QCQPs; treatments of even greater generality are also available [65, 67, 68]. Note that if an update is required for the example $(\mathbf{x}_t, y_t)$, it follows (by continuity) that the solution to eq. (17) will satisfy the margin constraint with equality. We therefore start by forming the Lagrangian

$$\mathcal{L}(\mathbf{U}, \mathbf{V}, \lambda) \ = \ \|\mathbf{U} - \mathbf{U}_t\|_F^2 + \|\mathbf{V} - \mathbf{V}_t\|_F^2 \ + \ \lambda \left( y_t - \|\mathbf{U}\mathbf{z}_t\|^2 + \|\mathbf{V}\mathbf{z}_t\|^2 \right). \tag{21}$$

The next step is to eliminate the matrices $\mathbf{U}$ and $\mathbf{V}$. By setting their partial derivatives to zero, we find that these matrices satisfy

$$\mathbf{U} \ = \ \mathbf{U}_t \left( \mathbf{I} - \lambda \mathbf{z}_t \mathbf{z}_t^\top \right)^{-1} \ = \ \mathbf{U}_t \left( \mathbf{I} + \frac{\lambda \mathbf{z}_t \mathbf{z}_t^\top}{1 - \lambda \|\mathbf{z}_t\|^2} \right), \tag{22}$$

$$\mathbf{V} \ = \ \mathbf{V}_t \left( \mathbf{I} + \lambda \mathbf{z}_t \mathbf{z}_t^\top \right)^{-1} \ = \ \mathbf{V}_t \left( \mathbf{I} - \frac{\lambda \mathbf{z}_t \mathbf{z}_t^\top}{1 + \lambda \|\mathbf{z}_t\|^2} \right). \tag{23}$$

To obtain the rightmost expressions in eq. (22–23), we have inverted the matrices $\mathbf{I} \pm \lambda \mathbf{z}_t \mathbf{z}_t$ using the Sherman-Morrison formula and assuming that $|\lambda| \neq \|\mathbf{z}\|^{-2}$. (We will verify this assumption in the course of finding a solution.) Given these expressions for $\mathbf{U}$ and $\mathbf{V}$, it remains only to determine the Lagrange multiplier $\lambda$. To do so, we simply enforce the constraint $y_t = \|\mathbf{U}\mathbf{z}_t\|^2 - \|\mathbf{V}\mathbf{z}_t\|^2$. Substituting eqs. (22–23) into this constraint, we obtain the nonlinear equation:

$$y_t \ = \ \left( \frac{1}{1 - \lambda \|\mathbf{z}_t\|^2} \right)^2 \|\mathbf{U}_t \mathbf{z}_t\|^2 - \left( \frac{1}{1 + \lambda \|\mathbf{z}_t\|^2} \right)^2 \|\mathbf{V}_t \mathbf{z}_t\|^2. \tag{24}$$

To simplify what follows, we make the elementary change of variables $\nu = y_t \lambda \|\mathbf{z}_t\|^2$. With this change, we can rewrite eq. (24) as

$$0 \ = \ \frac{d}{d\nu} \left[ \left( \frac{1}{1 - y_t \nu} \right) \|\mathbf{U}_t \mathbf{z}_t\|^2 + \left( \frac{1}{1 + y_t \nu} \right) \|\mathbf{V}_t \mathbf{z}_t\|^2 - \nu \right]. \tag{25}$$

Finally we observe that the derivative in eq. (25) is negative at $\nu = 0$ and vanishes for some $\nu \in (0, 1)$ when $y_t \left( \|\mathbf{U}_t \mathbf{z}_t\|^2 - \|\mathbf{V}_t \mathbf{z}_t\|^2 \right) < 1$; this inequality is, of course, precisely what triggers the update. In this way, we recover the minimization prescribed by eq. (18). Likewise, we obtain the passive-aggressive updates in eqs. (19–20) by substituting this solution for the step size into eqs. (22–23).

## 3.4 Model averaging

Passive-aggressive updates will not converge to a stable solution if there is no model that perfectly classifies the training examples, and they may converge very slowly even when there is. This may not matter if the model is to be indefinitely deployed in an online setting (e.g., when the examples are drawn from a nonstationary or adversarial distribution [69]). But it does matter if the ultimate goal is to train and deploy a single fixed model.

For linear classifiers, it is known that a stable result can be obtained by averaging models across time [28, 29, 70]. Let $\boldsymbol{\Theta}_m$ denote the model parameters obtained after $m$ intervals of training (where each interval corresponds to some period of natural interest—a single update, an epoch, a second, a minute, etc). In the online setting, an averaged model $\hat{\boldsymbol{\Theta}}_m = \frac{1}{m} \sum_{k=1}^m \boldsymbol{\Theta}_k$ can be computed at the end of each interval. This type of averaging has a simple intuition for linear models where the parameters specify a separating hyperplane. For example, consider two models whose separating hyperplanes are parallel. Averaging their parameters, we obtain another model whose separating hyperplane is sandwiched in the middle. For linear classifiers, this is a sensible result.

The situation is more complicated for *over-parameterized* models, and a simple example shows why. Consider two DoS models with parameters $\boldsymbol{\Theta}_+ = (\mathbf{U}, \mathbf{V})$ and $\boldsymbol{\Theta}_- = (-\mathbf{U}, -\mathbf{V})$; also consider the model with the averaged parameter $\hat{\boldsymbol{\Theta}} = \frac{1}{2}(\boldsymbol{\Theta}_+ + \boldsymbol{\Theta}_-)$. Note that the models $\boldsymbol{\Theta}_+$ and $\boldsymbol{\Theta}_-$ specify exactly the *same* classifier, whereas the model $\hat{\boldsymbol{\Theta}} = (\mathbf{0}, \mathbf{0})$ does not specify a decision boundary at all. In this case, the naive average has clearly not yielded a sensible result. What has gone wrong?

The problem arises in large part due to the symmetries of the model's parameter space. The continuous symmetries, in particular, define low-dimensional manifolds of functionally equivalent classifiers, and these manifolds are not convex. This lack of convexity can lead to nonsensical results when

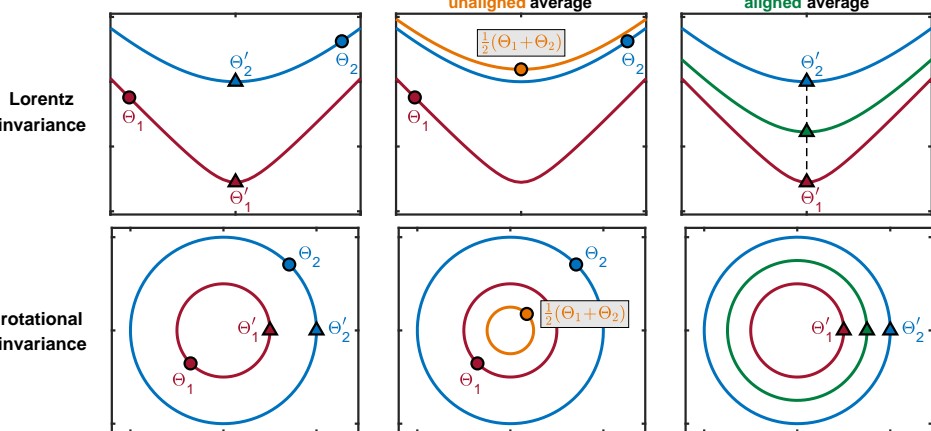

Figure 2: The DoS model is over-parameterized in the sense that different parameters $\boldsymbol{\Theta}$ and $\boldsymbol{\Theta}'$ can generate classifiers with the same decision boundary and confidence scores. This is most evidently true when $\boldsymbol{\Theta}$ and $\boldsymbol{\Theta}'$ are related by a continuous symmetry of the model's parameter space, such as a Lorentz transformation (*top*) or a rotation (*bottom*). Let $\boldsymbol{\Theta}_1, \boldsymbol{\Theta}'_1$ be one such pair of parameters, and let $\boldsymbol{\Theta}_2, \boldsymbol{\Theta}'_2$ be another (*left*). Suppose we desire a model that represents, in some meaningful sense, the *average* of the models defined by $\boldsymbol{\Theta}_1$ and $\boldsymbol{\Theta}_2$. Due to the model's symmetries, the midpoint $\frac{1}{2}(\boldsymbol{\Theta}_1 + \boldsymbol{\Theta}_2)$ may yield a nonsensical result for this average (*middle*). However, the midpoint $\frac{1}{2}(\boldsymbol{\Theta}'_1 + \boldsymbol{\Theta}'_2)$ will define a sensible average if $\boldsymbol{\Theta}'_1$ and $\boldsymbol{\Theta}'_2$ are appropriately aligned (*right*).

intermediate models are constructed by linear interpolation. The problem is illustrated in Fig. 2 for the particular symmetries of DoS models. The figure also depicts a natural solution to this problem: it is to *align* the parameters of different models before performing their average. Intuitively, by first aligning these parameters, we compensate for the invariances of the model's parameter space. The alignment can also be viewed as a form of continuous symmetry-breaking [60]; this symmetry-breaking is needed to compare or interpolate between different overparameterized models in a meaningful way.

More concretely, we perform this aligned average as follows. Suppose that we wish to compute a DoS model $\hat{\boldsymbol{\Theta}}$ that is intermediate between two other DoS models $\boldsymbol{\Theta}_1 = (\mathbf{U}_1, \mathbf{V}_1)$ and $\boldsymbol{\Theta}_2 = (\mathbf{U}_2, \mathbf{V}_2)$. We do this in three steps:

1. Account for the model's invariance under Lorentz transformations. In particular, compute the boost $\varphi_1$ in eq. (15) that yields the minimum-norm model $\boldsymbol{\Theta}'_1 = (\mathbf{U}'_1, \mathbf{V}'_1)$ equivalent to $\boldsymbol{\Theta}_1 = (\mathbf{U}_1, \mathbf{V}_1)$. Also compute the boost $\varphi_2$ that yields the minimum-norm model $\boldsymbol{\Theta}'_2 = (\mathbf{U}'_2, \mathbf{V}'_2)$ equivalent to $\boldsymbol{\Theta}_2 = (\mathbf{U}_2, \mathbf{V}_2)$.

2. Account for the model's invariance under orthogonal transformations. In particular, compute the matrices $\boldsymbol{\Omega}$ and $\boldsymbol{\Lambda}$ that solve the orthogonal Procrustes problems [71]:

$$\boldsymbol{\Omega} = \underset{\mathbf{Q}}{\operatorname{argmin}} \|\mathbf{Q}\mathbf{U}'_2 - \mathbf{U}'_1\|_F^2 \quad \text{such that} \quad \mathbf{Q}^\top \mathbf{Q} = \mathbf{I}, \tag{26}$$

$$\boldsymbol{\Lambda} = \underset{\mathbf{Q}}{\operatorname{argmin}} \|\mathbf{Q}\mathbf{V}'_2 - \mathbf{V}'_1\|_F^2 \quad \text{such that} \quad \mathbf{Q}^\top \mathbf{Q} = \mathbf{I}. \tag{27}$$

   Intuitively, $\boldsymbol{\Omega}$ and $\boldsymbol{\Lambda}$ are the orthogonal transformations that best align $\mathbf{U}'_2$ with $\mathbf{U}'_1$ and $\mathbf{V}'_2$ with $\mathbf{V}'_1$. These matrices have closed-form solutions that are easily obtained from the singular value decompositions (SVDs) [72] of the matrices $\mathbf{U}'_1, \mathbf{U}'_2, \mathbf{V}'_1$, and $\mathbf{V}'_2$. Set $\boldsymbol{\Theta}''_2 = (\boldsymbol{\Omega}\mathbf{U}'_2, \boldsymbol{\Lambda}\mathbf{V}'_2)$, and note that by construction the models $\boldsymbol{\Theta}''_2$ and $\boldsymbol{\Theta}_2$ define the same classifier.

3. Compute $\hat{\boldsymbol{\Theta}}$ by averaging the newly aligned parameters $\boldsymbol{\Theta}'_1$ and $\boldsymbol{\Theta}''_2$, as opposed to the original (unaligned) parameters: i.e., for an unweighted average, set $\hat{\boldsymbol{\Theta}} = \frac{1}{2}(\boldsymbol{\Theta}'_1 + \boldsymbol{\Theta}''_2)$.

We can also extend this procedure to average multiple DoS models across time. To do so, we note that the *unaligned* average $\hat{\boldsymbol{\Theta}}_m = \frac{1}{m}\sum_{k=1}^m \boldsymbol{\Theta}_k$ may be computed incrementally by setting

$$\hat{\boldsymbol{\Theta}}_m = \left(1 - \tfrac{1}{m}\right)\hat{\boldsymbol{\Theta}}_{m-1} + \tfrac{1}{m}\boldsymbol{\Theta}_m. \tag{28}$$

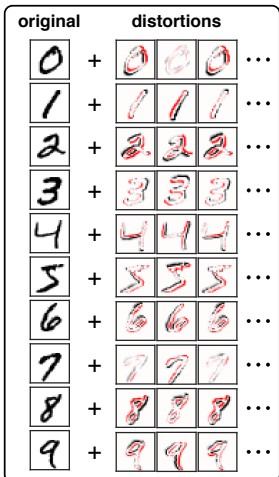 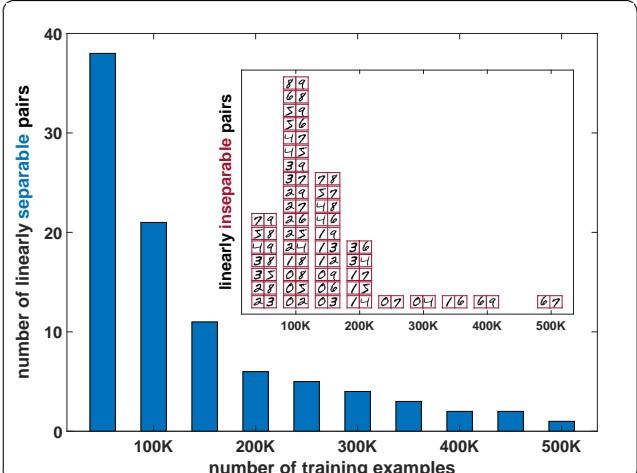

Figure 3: *Left*: The `INFIMNIST` data set [7] generates new digit images by distorting training examples from the original `MNIST` data set [73]. *Right:* Fewer pairs of digits remain linearly separable as more examples are added to the training set. The inset shows when (and which) pairs of digits become linearly inseparable as a function of the number of training examples.

For the *aligned* average, we simply replace the terms $\hat{\Theta}_{m-1}$ and $\Theta_m$ in eq. (28) by their equivalent but aligned counterparts $\hat{\Theta}'_{m-1}$ and $\Theta''_m$ from the above procedure.

The additional computations for these aligned averages do not entail much extra cost. The more costly step is the one that accounts for the model's invariance under orthogonal transformations. However, the cost of this step is small for two reasons. First, the model's parameter matrices are typically short and wide; thus in solving the orthogonal Procrustes problems of eqs. (26–27), it is fairly cheap to compute and perform SVDs of the $d \times d$ matrices $\mathbf{U}'_1 \mathbf{U}'^\top_2$ and $\mathbf{V}'_1 \mathbf{V}'^\top_2$. Second, in order to obtain a stable result, it is not necessary to average over *all* of the models estimated throughout the course of training. In practice, this average can be performed over the models estimated at regular but much longer intervals (e.g., once per several thousand training examples).

## 4 Experimental results

We experimented on the `INFIMNIST` data set of handwritten digit images [7], a purposefully constructed superset of the original `MNIST` data set [73]. The first 10K images of this superset are those of the `MNIST` test set, and the next 60K images are those of the `MNIST` training set. The remaining images—about one trillion of them—were programmatically generated by distorting the 60K images of the `MNIST` training set; see Fig. 3 (*left*). Our experiments set aside the first 10K images as a test set and used the next 100M images (without reshuffling) as training examples for online learning.

The `MNIST` data has been extensively benchmarked. For the original 60K training examples, it is well known that most pairs of digit classes are linearly separable. This situation changes, however, when the training data is augmented with more examples as described above. Fig. 3 (*right*) shows the number of linearly separable pairs of digits as a function of the number of `INFIMNIST` training examples. Note that only only one pair of digits (0 versus 1) remains linearly separable when the training set contains as few as 500K examples. This suggests a role for models of higher capacity.

Our experimental setup[2] was straightforward. We trained linear and `DoS` models using the updates, respectively, from eqs. (2) and (19–20). We initialized the parameters of the linear model with zero values and those of the `DoS` models with small random values. Specifically, we sampled the elements of $\mathbf{U}$ and $\mathbf{V}$ from a zero-mean normal distribution whose variance was inversely proportional to the number of elements in these matrices. (Note that the `DoS` models require a nonzero initialization to break the symmetry between different hidden units.) We computed ten-way error rates from the

---

[2]In total we purchased several hundred CPU-hours on an externally managed cluster of Intel Xeon Gold 6132 processors. Much of this was expended on prototyping, debugging, and other intermediate or non-results.

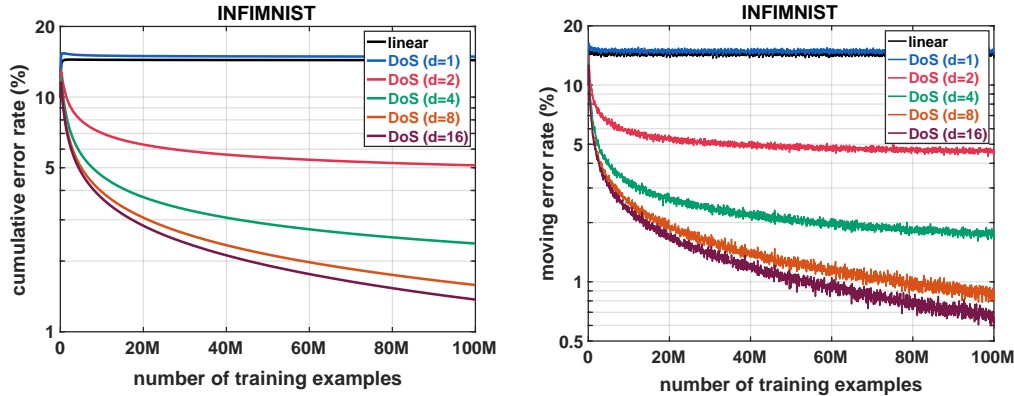

Figure 4: Cumulative and moving error rates (%) on training examples for different models of `INFIMNIST` digits. The latter were computed on non-overlapping sliding windows of 50K examples.

majority votes of $45 = \frac{1}{2}(10 \cdot 9)$ pairwise classifiers. There were no other sources of randomness in the experiments other than the seed for the random number generator. Though not shown here, we verified on the first 10M examples that the results varied little across five different values of this seed.

Fig. 4 compares the cumulative and moving error rates on training examples from different models. The moving error rates were computed on non-overlapping sliding windows of 50K examples. The results are shown for the linear model as well as `DoS` models with hidden vectors $\mathbf{h}_\pm$ of dimensionality $d \in \{1, 2, 4, 8, 16\}$. It is clear that the `DoS` models with higher capacity learn more accurate classifiers. In addition, as one might expect, the error rates stabilize more quickly for smaller models. For the largest model, the error rates are still decreasing after 100M examples, showing that even at this stage they are still improving.

Fig. 5 compares the *test* error rates from models sampled at intervals of 50K training examples. The error rates of non-averaged models (*left*) are extremely variable, especially for the smaller models. However, the results stabilize quickly when these models are averaged across time (*right*). In addition, it is clear that the averaged models (*right*) generalize much better than the non-averaged ones (*left*). It required some extra computation in these experiments to average different `DoS` models (see section 3.4), but these averaging steps were only performed once per 50K training examples. Thus the extra cost was essentially negligible.

Overall the results show that `DoS` models of low rank can learn much more accurate classifiers than linear models. In addition, these improvements are obtained from online updates that are nearly as simple to implement; compare eqs. (19–20) to eq. (2). Our results provide further evidence for the benefits of low-rank models of quadratic classification, and they are consistent with previous results, on the original MNIST data set, that factorization machines can learn much more accurate classifiers than purely linear models [74]. This paper is the first, however, to demonstrate the viability of passive-aggressive updates for these types of models.

It is curious that the linear model slightly outperforms the `DoS` model with rank-one ($d=1$) matrices, even though the latter contains the former as a special case. We do not entirely understand this effect, but we did observe it repeatedly in our experiments. Presumably this performance gap can be traced to the different objective functions for these models. Both the linear and `DoS` models can be viewed as attempting to minimize an objective function based on the sum of hinge losses. For the linear model, however, this objective function is convex in the model parameters, while for the `DoS` models, it is not. It seems that `DoS` models with larger capacity easily overcome[3] this non-convexity to surpass the linear model, while the `DoS` model with rank-one matrices is not able to do so. It should be kept in mind, though, that the performance gap between the linear and rank-one `DoS` model is far smaller than the gap between both these models and `DoS` models of higher rank.

---

[3]Analogous behavior has been observed in neural network classifiers: those with barely more capacity than a linear model may not always improve on (say) logistic regression, while those with large numbers of hidden units do so without any difficulty.

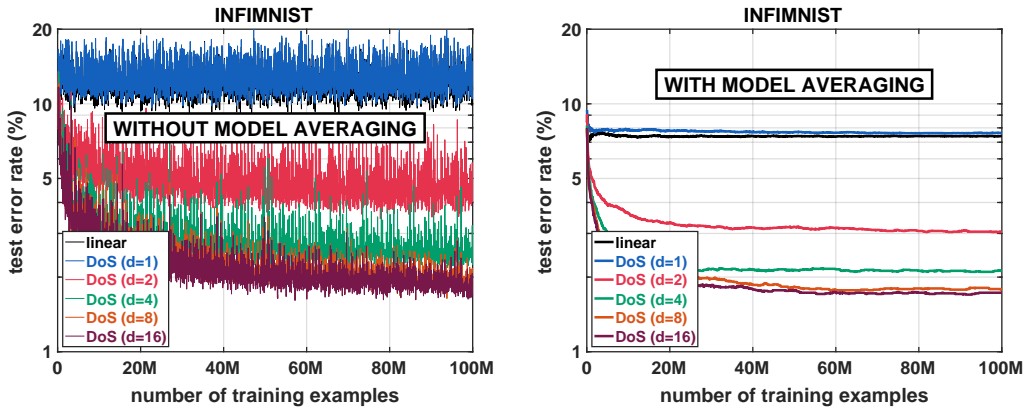

Figure 5: Test error rates (%) for different models of INFIMNIST digits as a function of the number of training examples. The models on the right were averaged across time; those on the left were not.

## 5  Conclusion

In this paper we have investigated a family of low-rank models for quadratic classification. Like factorization machines and polynomial networks, these models provide a natural bridge between purely linear and fully quadratic classifiers. The DoS models in this paper were trained by passive-aggressive updates; these updates do not require the tuning of learning rates, and they are only slightly more complicated than the passive-aggressive updates already in use for linear models. For all of these reasons, the models in this paper should provide a useful addition to the practitioner's toolbox.

These models also raise several questions that are deserving of further theoretical study. Many formal results are available for passive-aggressive learning of linear classifiers, including regret bounds on the cumulative number of mistakes [3]. It remains to show whether these bounds can be transferred to DoS classifiers. It would also be interesting to derive some sort of (*Lorentztron?*) convergence theorem for the realizable setting.

In pursuing this work, we realize that any technological advance carries the risk of negative societal impacts. There may be settings (e.g., battlefields, markets, networks) where better online algorithms could be exploited by adversaries to overcome static or slowly evolving defenses. More generally, any classifier of sensitive data can be misapplied, deployed with bias, or deliberately put to misuse [77–79]. By raising awareness of these negative impacts, however, we can hope to forestall them.

We conclude by observing that the fields of machine learning and optimization are inextricably linked. The simplest linear models [80–82] are powered by least-squares methods and singular value decompositions [72]. With extra machinery, these models can be made more robust [83–85] or interpretable [86] or even extended to nonlinear settings [87]; these generalizations are often formulated as convex quadratic or semidefinite programs [88]. In machine learning as a whole, optimizations that once seemed exotic are now commonplace. In the study of passive-aggressive learning algorithms, this paper has found a role for nonconvex but tractable QCQPs. It seems likely that other useful models, yet to be formulated, can benefit from these methods.

### Acknowledgements

We are grateful to the anonymous reviewers and NeurIPS area chairs for numerous suggestions to improve this manuscript. The paper's experiments were performed on the Triton Shared Computing Cluster (TSCC) at UC San Diego and supported by an allowance from the author's home department.

### Funding Transparency Statement

There was no funding in direct support of this work, and the author did not have financial relationships with entities that could be perceived to influence the contents of this paper.

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
