# OpenReview forum: "An online passive-aggressive algorithm for difference-of-squares classification"
_NeurIPS.cc/2021/Conference — NeurIPS 2021 Poster_

### Official Review · Reviewer_Tg7a · 2021-07-12

**Rating:** 7
**Confidence:** 4

**Summary:**

In the setting of online learning, this paper investigates difference-of-squares models with the aim of devising conceptually simple and efficient passive-aggressive algorithms. Starting from the background on online passive-aggressive linear classification, the authors present the difference-of-squares model, by highlighting its symmetries. For this model, the update step and the averaging step are justified in detail. Experimental results on the InfiMinst dataset corroborate the interest of this quadratic passive-aggressive approach, especially in comparison with the standard, linear approach.


**Ethical Concerns:**

Not Applicable.

**Limitations And Societal Impact:**

Yes, most of the limitations of this study have been pointed out by the authors. Societal impacts are not applicable in the present context.

**Main Review:**

Overall, this paper is very well-written, well-motivated, and well-organized. The key issues related to the nonconvexity and over-parameterization of difference-of-squares models are examined in detail, and the solutions are elegant. Finally, the (preliminary) experimental results look promising.

I have only minor comments, in order to further improve this paper:
* In the Supplementary Material, it would be nice to provide more details about the “S-procedure” used to solve the nonconvex QCQP problem (16).
* At the end of Section 3.4, it would be relevant to give a brief comment about the runtime complexity of the three-step model averaging method (I think that it is dominated by the computation of singular value decompositions).
* As indicated by the authors, the lack of regret analysis is a limitation. To this point, I think that they are two main directions of research: mistake-bounds in the realizable case (i.e. the “Lorentztron” analysis), and regret-bounds in the agnostic case (involving the choice of an appropriate surrogate loss for such difference-of-squares models).


**Time Spent Reviewing:**

3

---

> ### Author Response · Authors · 2021-08-09
> **Clarification of averaging steps**
>
> Thank you for your careful reading of this paper, as well as the detailed comments and helpful suggestions for improvement.
>
> You asked specifically about the cost of the averaging method. The computation time (as you intuited) is dominated by the singular value decompositions of two (n+1) x d matrices, where n is the input dimensionality and d is the number of hidden units per class. For small d (as in our experiments, where d<=16),  these SVDs are very fast. But even more importantly, it is *not* necessary to perform these SVDs after every online update. In particular, we only performed an averaging step after every 50000 updates. Thus the total cost of the averaging steps was essentially negligible. We can clarify this in a revised manuscript.
>
> You suggest some promising directions for future work on theoretical results. We agree, and we hope to entice the computational learning theorists at NeurIPS to pursue these directions.

---

### Official Review · Reviewer_HDQ3 · 2021-07-16

**Rating:** 5
**Confidence:** 3

**Summary:**

The paper investigates a family of low-rank models for quadratic classification inspired by so-called capsule networks for visual object recognition. For models in this family, it shows how to derive passive-aggressive updates by solving a quadratically constrained quadratic program (QCQP).

**Limitations And Societal Impact:**

See my comments in "Main Review".

**Main Review:**

I am not an expert on online learning so I cannot evaluate the significance of this work. But the methods proposed here look useful to me.  A potential drawback of this work is that the authors only validate their method on a single dataset. Therefore, I am not sure the approach proposed here generalizes well to other datasets. I, therefore, refer to other reviewers on the evaluation of the significance of this work.

**Time Spent Reviewing:**

1

---

> ### Author Response · Authors · 2021-08-09
> **Thank you**
>
> Thank you for your comments. It is heartening to read that the proposed methods are accessible (and seem useful) to a broad audience.

---

### Official Review · Reviewer_15ax · 2021-07-19

**Rating:** 6
**Confidence:** 4

**Summary:**

The authors proposed a low rank model for quadratic classification formulated as a neural network, which they called Difference-of-Squares (DoS). Albeit not a universal approximator, such as neural networks, the proposed model can compete with any fully quadratic classifier. The ablation study involved using the INFIMNIST dataset and the DoS model was compared against a linear classifier. There has also been a bizarre outcome presented, i.e. linear model outperforming the DoS with dimensionality=1. Many thanks to the authors for mentioning this and being honest that they do not know why this is happening.

**Limitations And Societal Impact:**

I have mentioned the limitations above. I think they mainly refer to the experimental setup, conclusion/discussion and mentioning capsule networks but not being more specific.

**Main Review:**

The DoS model authors presented is based on a neural net formulation albeit "constrained" to a quadratic classification. The mathematical formulation of the problem has been robust and detailed and has clearly reflected upon the symmetries of the model in that the decision boundary and confidence scores only depend on its parameters through the difference U^TU-V^TV. This implies that the model is invariant to orthogonal transformations and also invariant to Lorentz transformations.
Furthermore, the learning process considers a passive-aggressive updates for online learning whereby the labelled examples arrive one at a time. The purpose is the model parameters/weights to be updated at a Δ that is enough to classify the initially wrongly classified point.

Although the approach is convincing the experimental setup is slightly lacking, not only in terms of datasets considered but on how it has been presented. For instance, would worse results for d=1 for DoS be evidenced across other datasets too? This would have allowed to check the consistency of the outputs beyond one dataset (this is just a limitation not a deal-breaker).

Perhaps a table would have helped to summarise the main findings and be more reflective on the importance of the findings. I think the conclusion could be split into another section "Discussion" as it is currently too long and it contains discussion either way.

Lastly, I was hooked initially that the paper will have some associations with Capsule Networks, but I ended up wondering, what has the purpose been of mentioning them? Just their viewpoint invariant capabilities?

**Time Spent Reviewing:**

2 hours

---

> ### Author Response · Authors · 2021-08-09
> **further discussion of d=1 results and connection to capsule networks**
>
> Thank you for your careful reading of this paper, as well as the detailed comments and helpful suggestions for improvement.
>
> After further study, we can offer a simple intuition why the DoS model with d=1 performs slightly worse than the linear model (despite the fact that the former has a larger model capacity). Intuitively, both the linear and DoS classifiers can be viewed as attempting to minimize an objective function based on the sum of hinge losses. Note that for the linear classifier, this objective function is convex, while for the DoS classifiers, it is not. It seems that the DoS models with much larger capacity easily overcome this non-convexity to surpass the linear model, while the DoS model with d=1 is not able to do so. One observes analogous behavior in vanilla neural networks: those with barely more capacity than a linear model do not necessarily improve on logistic regression or linear autoencoders, while those with large numbers of hidden units do so without any difficulty. So perhaps this behavior in the DoS model is not so bizarre.
>
> (It should also be kept in mind that the difference between the linear and d=1 DoS model is absolutely minuscule compared to the amount by which both these models are outperformed by DoS models of higher rank.)
>
> The paper can and should do a better job at making the connection to capsule networks. In spite of this, you seem to have grasped the essential connection, which is that in capsule networks, the probability for detecting an object part is determined by the vector magnitude of a capsule's hidden unit activities. These magnitudes are rotationally invariant, and this is the same rotational invariance that is observed in DoS models. Likewise, the hidden units in DoS models can be viewed as encoding viewpoint/pose information for each class.

---

### Official Review · Reviewer_ymSV · 2021-07-26

**Rating:** 3
**Confidence:** 4

**Summary:**

Passive Aggressive Algorithms are a very important class for algorithms for large-scale learning. This paper makes some modeling contributions.

**Ethical Concerns:**

There are no ethical concerns.

**Limitations And Societal Impact:**

Yes

**Main Review:**

Passive Aggressive Algorithms are a very important class for algorithms for large-scale learning. These are not very popular but recently there has been a lot of activity to improve and extend these models. The present paper extends these ideas to nonlinear models without taking recourse to Kernels. Instead, their work is motivated by capsule networks for visual object recognition. I liked the motivation and problem formulation. While the authors claim that their work contributes to understanding low-rank models for quadratic classification and provide a "natural bridge between purely linear and fully quadratic classifiers", I don't see why this work is significant in the absence of any theoretical results. Further experimental results are very limited and unless otherwise, they compare results with other online algorithms it is very difficult to make sense of these. Also, the computational costs involved in this too not very clear. I should say that the passive-aggressive update that the authors derived is pretty neat. But so what? After all this, there are no theoretical insights? At least some bounds on the cumulative loss of the algorithms?

**Time Spent Reviewing:**

4.5 hours

---

> ### Author Response · Authors · 2021-08-09
> **Response to reviewer comments**
>
> Thank you for your comments.
>
> * You state that the experimental results are very limited and that the results are not compared with other online algorithms.
>
> 1. The models in the paper were benchmarked on a data set with 100 million examples.
>
> 2. These models were explicitly compared to a linear classifier with passive-aggressive updates, which is the relevant baseline.
>
> * You state that the computational costs are not clear.
>
> 1. The updates in this paper are nearly as easy to implement as those for linear models, and the paper states this explicitly.
>
> 2. The scalability is demonstrated by experimental results on a data set with 100 million examples.
>
> 3. Could you be more specific about the algorithm's perceived costs? (I.e., to which equations are you referring?)
>
> * You ask how a work can be significant in the absence of any theoretical results.
>
> 1. The paper has several mathematical results that are not stated as theorems (e.g., the solution of a nonconvex QCQP, the characterization of symmetries, the Lorentz transformation for a model of minimum norm).
>
> 2. An algorithm can be significant in the absence of theoretical results if it is elegant, effective, scalable, and easy to implement.
>
> 3. Passive-aggressive algorithms have been widely studied by computational learning theorists who prove regret bounds for linear or kernel-based classifiers. Perhaps a somewhat different perspective on these algorithms may help to expand their reach.

---

> > ### Comment · Reviewer_ymSV · 2021-08-31
> > **Thanks you for you replies.**
> >
> > Thank you for the replies. I strongly feel that some bounds on cumulative loss would immenstly strengthen the paper.

---

### Decision · Program_Chairs · 2021-09-27

**Decision:**

Accept (Poster)

**Comment:**

The paper introduces a new  passive-aggressive algorithms where the hypothesis function is a difference of squares (quadratics). Previously, only linear models had been considered with the passive-aggressive framework. The resulting model is non-convex, but tractable, since the authors show the model parameters can be updated using a 1-dimensional convex optimization problem over the interval (0, 1).  The authors then study the symmetries in the resulting model, and develop a specialized averaging scheme which is non-trivial given nonlinear manifold for which the model presents. These are the strengths of the paper, and I believe that the ideas present in the paper on using difference of squares models, extending the passive-aggressive framework to new model and averaging will be useful and interesting to the community.

The paper has some weakness for which I would like to see the authors make some small adjustments for the camera ready:
1) A lack of mention of comparison to other quadratic based models. As such, I would kindly request the authors to also mention factorization machines, and other related quadratic models.
2) Insufficient numerics. Currently the numerics compare the DoS (difference of squares) model to a linear model, and ultimately show the DoS model is more expressive, without overfitting. There are also experiments highlighting the benefits on the new form of averaging. But  additional experiments that could disentangle the benefits of using DoS model, from the passive-aggressive framework, and also contrast to other quadratic models (such a factorization machines) would really enrich the paper. So I kindly request that the authors consider including some additional numeric insights along these lines in the camera ready paper.
3)  The authors also highlight how the DoS model can interpolate data, and thus this could be used as a simple case study for the interpolation of DNNs. But no details, or hints as to how or why this makes sense were developed. Perhaps the authors could tone down this claim?

Finally, the reviews where divided on the paper, but ultimately the reviewers in favor raised more concrete points, which after my own reading of the paper I agreed with.